# Evaluation of Effect of Brassinolide in *Brassica juncea* Leaves under Drought Stress in Field Conditions

Naveen Naveen [1], Nisha Kumari [1,*], Ram Avtar [2], Minakshi Jattan [3], Sushil Ahlawat [4], Babita Rani [1], Kamla Malik [5], Anubhuti Sharma [6] and Manjeet Singh [2,*]

[1] Department of Biochemistry, CCS Haryana Agricultural University, Hisar 125004, Haryana, India; jangranaveen4321@gmail.com (N.N.); babitachahalkharb@gmail.com (B.R.)
[2] Oilseeds Section, Department of Genetics and Plant Breeding, CCS Haryana Agricultural University, Hisar 125004, Haryana, India; ramavtar0706@gmail.com
[3] Cotton Section, Department of Genetics and Plant Breeding, CCS Haryana Agricultural University, Hisar 125004, Haryana, India; jattanmina@gmail.com
[4] Department of Entomology, CCS Haryana Agricultural University, Hisar 125004, Haryana, India; sushilahlawat08@gmail.com
[5] Department of Microbiology, CCS Haryana Agricultural University, Hisar 125004, Haryana, India; kamlamalik06@gmail.com
[6] ICAR-Directorate of Rapeseed-Mustard Research, Sewar, Bharatpur 321303, Rajasthan, India; sharmaanubhuti98@gmail.com
* Correspondence: nishaahlawat211@gmail.com (N.K.); manjeetsingh125033@gmail.com (M.S.); Tel.: +91-9468117723 (N.K.); +91-9053127937 (M.S.)

**Abstract:** Drought stress is considered to be a major factor responsible for reduced agricultural productivity, because it is often linked to other major abiotic stresses, such as salinity and heat stress. Understanding drought-tolerance mechanisms is important for crop improvement. Moreover, under drought conditions, it is possible that growth regulators are able to protect the plants. Brassinosteroids not only play a regulatory role in plant growth, but also organize defense mechanisms against various tresses. This study aimed to evaluate the effect of brassinolide on physio-biochemical amendment in two contrasting cultivars (drought-tolerant RH 725, and drought-sensitive RH 749) of *Brassica juncea* under drought stress. Two foliar sprayings with brassinolide (10 and 20 mg/L) were carried out in both cultivars (RH 725 and RH 749) at two stages—i.e., flower initiation, and 50% flowering—under stress conditions. The results clearly revealed that the activities of antioxidative enzymes and non-enzymatic antioxidants (carotenoids, ascorbic acid, and proline) increased significantly in RH 725 at 50% flowering, whereas 20 mg/L of brassinolide showed the most promising response. The different oxidative stress indicators (i.e., hydrogen peroxide, malondialdehyde, and electrolyte leakage) decreased to a significant extent at 20 mg/L of brassinolide spray in RH 725 at 50% flowering. This study indicates that brassinolide intensifies the physio-biochemical attributes by improving the antioxidant system and photosynthetic efficiency in RH 725 at 50% flowering. It is assumed that enhanced production of proline, improvement of the antioxidant system, and reduction in the amount of stress indicators impart strength to the plants to combat the stress conditions.

**Keywords:** antioxidants; *Brassica juncea*; brassinolide; drought stress; proline

## 1. Introduction

Rapeseed mustard comprises an important group of oilseed Brassica crops. In this group, Indian mustard [*Brassica juncea* (L.) Czern & Coss.] is an important edible, oil-yielding crop covering about 90% of the cultivated area under brassica oilseeds in India [1]. It is the third-largest source of vegetable oil in the world, after soybeans and palm oil. Indian mustard has the potential for quicker seed germination, high productivity, and heat and drought tolerance, along with enhanced insect and disease resistance if sown on time [2], whereas late sowing exposes the crop to abiotic and biotic stresses. There is a dire

need to intensify in the production of food crops but, on the other hand, environmental stresses (biotic and abiotic) suppress the overall yield of agricultural crops. Drought stress is recognized as the main factor leading to the decline in agricultural productivity, because drought is persistently related to other major abiotic stresses, such as high-temperature stress and salinity [3]. It is estimated that by the end of the 21st century, the proportion of drought-prone areas will have doubled.

Reactive oxygen species (ROS) are continuously generated in plant mitochondria, plastids, peroxisomes, apoplasts, and cytosol as byproducts of different cellular metabolic pathways, and they hinder photosynthesis. The enhanced production of alkoxy radicals ($RO^-$), superoxide radicals ($O^{2-}$), perhydroxy radicals ($HO^{2-}$), hydrogen peroxide ($H_2O_2$), singlet oxygen ($_1O^2$), and hydroxyl radicals ($OH^-$) is a common end result of plant-rearing under different abiotic stresses [4,5]. The production of reactive oxygen species is the basis for oxidative stress, damaging plants by oxidizing membrane lipids, nucleic acids, proteins, and photosynthetic pigments [6]. The ROS hamper the plants' photosynthesis and enzymes of the Calvin cycle, altering chlorophyll components and causing damage to the photosynthetic apparatus [7]. To survive under such intense environmental conditions, and to increase their tolerance, plants have developed many intricate defense mechanisms. Stress tolerance in plants necessitates the activation of complex metabolic activities, including antioxidative pathways—especially ROS-scavenging systems within the cells that, in turn, can contribute to continued plant growth under stress conditions [8]. The different antioxidative enzymes in plants—such as superoxide dismutase (SOD), catalase (CAT), and peroxidase (POX)—scavenge these ROS molecules [9]; however, oxidative stress is generated in plants if there is an imbalance in ROS [10]. The resistance to oxidative stress relies on the overall balance between the fabrication of ROS and the antioxidant capability of the cells [11].

The drought-tolerance mechanism in plants also includes some plant growth regulators and secondary metabolites, such as auxin, abscisic acid (ABA), jasmonic acid, plant steroids, and ethylene. Among the variety of compounds used to alleviate plant stress, brassinosteroids (BRs) are considered to be plant hormones that regulate plant growth and productivity. Brassinosteroids (BRs) are polyhydroxylated steroidal plant hormones that play an essential role in the regulation of plant growth and development processes. Myriad studies have highlighted that these are crucial for regulating a range of physiological processes, such as cell proliferation, expansion, male fertility, senescence, leaf development, and vascular differentiation. These compounds have a wide range of biological activities, providing unique possibilities to increase crop yields by altering plant metabolism and protecting plants from environmental stresses [12]. The research conducted thusfar shows that BRs cause a wide range of morphological and physiological responses in plants [13,14]. In addition, BRs are known as regulators of transcription and translation mechanisms, by which they improve the levels of total proteins and enzymes [15], as well as increasing the seed yield at harvest [16]. BRs not only play a regulatory role in plant growth, but also participate in the establishment of defense mechanisms to deal with various biotic and abiotic stresses [13]. Several BRs with brassinolide as the main component have been evaluated in the field, and they have significantly increased crop yields. Exogenous application of BRs has improved tolerance to salinity [17], drought [16,18], high/low temperatures, and heavy metals [13]. There are few reports on the role of brassinosteroids in the unveiling of genes and metabolic pathways that confer drought resistance to Indian mustard [16]. However, the data that are currently available on the role of BRs in plant drought response, from the few studies that have been performed with genotypes of known drought sensitivity, are not very conclusive [19–21]. Any comparison of the impact of exogenously applied BRs on drought-tolerant/sensitive genotypes should reveal the BR-induced changes—particularly in the sensitive genotypes, because the tolerant genotypes should experience less intensive drought effects. This should be similar to the situation observed for BRs exogenously applied to plants exposed to drought ranging from mild/moderate to less intense; BRs always have a greater effect on more strongly stressed plants. There have also been some

cases where the drought-tolerant genotype showed a more pronounced response to BRs than the drought-sensitive one, as has been reported in several previous studies. Thus, the situation is not so simple, and probably depends on plant species as well as on a mechanism that is responsible for the drought resistance/sensitivity of the respective genotype. In this context, the present study was designed with the objectives to address the following questions: (1) whether exogenous application of BRs could alleviate drought stress in Indian mustard, and (2) whether the drought-tolerant and drought-sensitive cultivars have similar responses to these treatments under drought stress.

## 2. Materials and Methods

### 2.1. Plant Materials

Two Indian mustard [*Brassica juncea* (L.) Czern & Coss.] cultivars—drought-tolerant (RH 725) and drought-sensitive (RH 749)—were used in this study. The two cultivars were sown in the Research Farm of Oilseeds Section, Department of Genetics and Plant Breeding, CCS Haryana Agricultural University, Hisar, in a randomized complete block design (RCBD) with three replications. The CCS HAU, Hisar, is situated at a latitude of 29°10′ N, longitude of 75°46′ E, and altitude 215.2 m above main sea level, and falls in the semi-tropical region of the western zone of India. Drought conditions were achieved by withholding irrigation from the crop. The weather data (rainfall) during crop season are presented in Supplementary Table S1, which shows that rainfall was negligible during the crop-growing period and the drought conditions were adequate for this study. Two foliar sprayings of brassinolide at 10 and 20 mg/L concentrations were carried out in both of the cultivars at two growth stages—i.e., flower initiation (42 days after sowing; DAS) and 50% flowering (52 DAS)—with water spray as a control. All physio-biochemical analysis was carried out on leaves, which were taken two days after each spray.

### 2.2. Physiological Parameters

Parameters such as photosynthetic rate, stomatal conductance, and transpiration rate were measured using an infrared gas analyzer (IRGA) system (LI-COR USA Model LI6400, LE, USA) as per the method employed by Silva et al. [22], and their corresponding units are as follows: photosynthetic rate (PR, $\mu$mol $CO_2$ $m^{-2}$ $s^{-1}$), stomatal conductance (SC, mole $H_2O$ $m^{-2}$ $s^{-1}$), and transpiration rate (TR, mmol $H_2O$ $m^{-2}$ $s^{-1}$).

### 2.3. Enzyme Extraction and Assay

Leaves of both cultivars of *B. juncea*, after two days of each spray, were used for enzymatic studies. Extraction was carried out at 4 °C, and standardized extraction conditions with respect to the molarity and pH of the buffer were maintained in order to achieve maximum enzyme activity. One gram of leaf sample was macerated in a chilled pestle and mortar in the presence of 4 mL of 0.1 M potassium phosphate buffer (pH 7.0). The homogenate was centrifuged at 12,000× $g$ rpm for 30 min in a refrigerated centrifuge at 4 °C. The supernatant was carefully decanted and used for the enzyme assay.

The SOD activity was determined by quantifying the ability of the enzyme to inhibit the photochemical reduction of nitro blue tetrazolium (NBT) to formazan [23]. One enzyme unit was defined as the amount of enzyme that could cause 50% inhibition of the photochemical reaction. The catalase activity was measured by following the method of Sinha [24]. One unit of enzyme activity was defined as the amount of enzyme required to consume 1 $\mu$mol $H_2O_2$ per minute under assay conditions. The POX activity was assayed by adopting the method of Shannon et al. [25]. One unit of enzyme activity was equivalent to one $\mu$mol of $H_2O_2$ oxidized per minute.

### 2.4. Non-Enzymatic Estimations

For extraction of carotenoids, 30 mg of the fresh leaves was cut into small discs and dipped in test tubes containing 3 mL of dimethyl sulfoxide (DMSO). The tubes were kept at room temperature overnight. The carotenoids extracted in the DMSO were estimated by

the method of Hiscox and Israelstam [26]. Ascorbic acid was extracted from the leaves via homogenization in 5 mL of 5% (*w/v*) metaphosphoric acid in glacial acetic acid, and the homogenate was centrifuged at $10,000 \times g$ rpm for 25 min. The supernatant thus obtained was used for the estimation of ascorbic acid. Ascorbic acid content was estimated by the method of Roe [27]. For proline estimation, 1g of tissue was homogenized in 5 mL of sulfosalicylic acid (3%) and centrifuged at $10,000 \times g$ rpm for 25 min; the supernatant thus obtained was used for the estimation of proline content. Proline content was estimated by using the method of Bates et al. [28].

### 2.5. Extraction and Estimation of Oxidative Stress Indicators

For the extraction of $H_2O_2$ and MDA, 1.0 g of leaves from each treatment was taken and ground in 6 mL of chilled 0.8 N $HClO_4$ and centrifuged at $10,000 \times g$ rpm for 30 min. The clear supernatant thus obtained was used for further estimation. Hydrogen peroxide was estimated by the method of Sinha [24]. Malondialdehyde was estimated according to the method of Heath and Packer [29].The relative intactness of the plasma membrane was measured as the leakage percentage of electrolytes, as described by Gong et al. [30].

### 2.6. Statistical Analysis

Three-way ANOVA was applied to test the statistical significance of the treatments. Duncan'smultiple range test (DMRT) was applied for multiple comparisons of treatments' mean values. Pearson's product–moment correlation was used to test the relationships between the antioxidant parameters. All statistical analyses were performed using the OP STAT statistical software developed by CCS HAU, Hisar, India. Graphs were prepared using Microsoft Excel, 2013.

## 3. Results

### 3.1. Physiological Parameters

The stomatal conductance, photosynthetic rate, and transpiration rate were significantly influenced by cultivar, growth stage, brassinolide concentration, and their interactions (Table 1). The brassinolide foliar application (10 and 20 mg/L) increased the stomatal conductance, photosynthetic rate, and transpiration rate, and these increases had the greatest statistical significance in the drought-tolerant cultivar RH 725 as compared to drought-sensitive RH 749 at both the growth stage and the 50% flowering stage. The photosynthetic parameters at the 50% flowering stage were high with brassinolide (10 and 20 mg/L) spray in both cultivars (RH 725 and RH 749). Brassinolide (20 mg/L) enhanced the stomatal conductance, photosynthetic rate, and transpiration rate by 47.16, 40.92, and 31.06% at the flower initiation stage and 50.72, 46.04, and 33.76% at the 50% flowering stage, respectively, in RH 725. This enhancement was low in drought-sensitive RH 749 as comparison to drought-tolerant RH 725 (Figures 1–3).

**Table 1.** Analysis of variance in the effects of cultivars (C), sampling time points (ST), and different brassinolide concentrations (BC),as well as their interactions, on the superoxide dismutase activity (SOD), peroxidase activity (POX), catalase activity (CAT), ascorbic acid (ASA), proline (PRO), carotenoids (CC), hydrogen peroxide ($H_2O_2$), and malondialdehyde (MDA) content, as well as electrolyte leakage (EL), stomatal conductance (SC), photosynthetic rate (PR), and transpiration rate (TR).

| SV | Df | Mean Squares | | | | | | | | | | | |
|---|---|---|---|---|---|---|---|---|---|---|---|---|---|
| | | SOD | POX | CAT | ASA | PRO | CC | $H_2O_2$ | MDA | EL | SC | PR | TR |
| C | 1 | 1335.17 ** | 61.96 ** | 27,408.11 ** | 1333.49 ** | 26.87 ** | 22.14 ** | 38,037.56 ** | 59.50 ** | 225.91 ** | 0.65 ** | 147.87 ** | 5.48 ** |
| ST | 1 | 1400.01 ** | 18.30 ** | 6702.90 ** | 2295.71 ** | 46.10 ** | 18.16 ** | 24,628.80 ** | 44.94 ** | 2334.83 ** | 0.29 ** | 98.01 ** | 59.78 ** |
| C × ST | 1 | 190.35 ** | 0.40 | 88.59 ** | 19.90 ** | 0.61 | 0.79 | 150.79 ** | 8.05 ** | 8.17 ** | 0.01 ** | 12.39 ** | 0.08 ** |
| BC | 2 | 137.58 ** | 31.86 ** | 5914.39 ** | 1068.52 ** | 11.94 ** | 7.58 ** | 18,979.63 ** | 178.78 ** | 242.52 ** | 0.14 ** | 98.06 ** | 6.72 ** |
| C × BC | 2 | 14.73 ** | 1.79 | 320.08 ** | 33.10 ** | 0.51 | 0.47 | 178.68 ** | 0.84 | 4.07 * | 0.02 ** | 8.20 ** | 0.20 ** |
| ST × BC | 2 | 15.35 ** | 0.48 | 192.53 ** | 6.57 ** | 0.63 | 0.48 | 124.60 ** | 5.13 ** | 1.26 | 0.01 ** | 2.13 ** | 0.54 ** |
| C × ST × BC | 2 | 1.95 * | 0.01 | 2.95 * | 0.47 | 0.00 | 0.12 | 94.83 ** | 1.09 | 0.29 | 0.00 | 0.46 ** | 0.06 ** |
| Error | 22 | 0.39 | 0.90 | 0.61 | 0.99 | 0.53 | 0.82 | 0.84 | 0.83 | 0.78 | 0.00 | 0.02 ** | 0.01 ** |

** Significant at $p \leq 0.01$; * significant at $p \leq 0.05$.

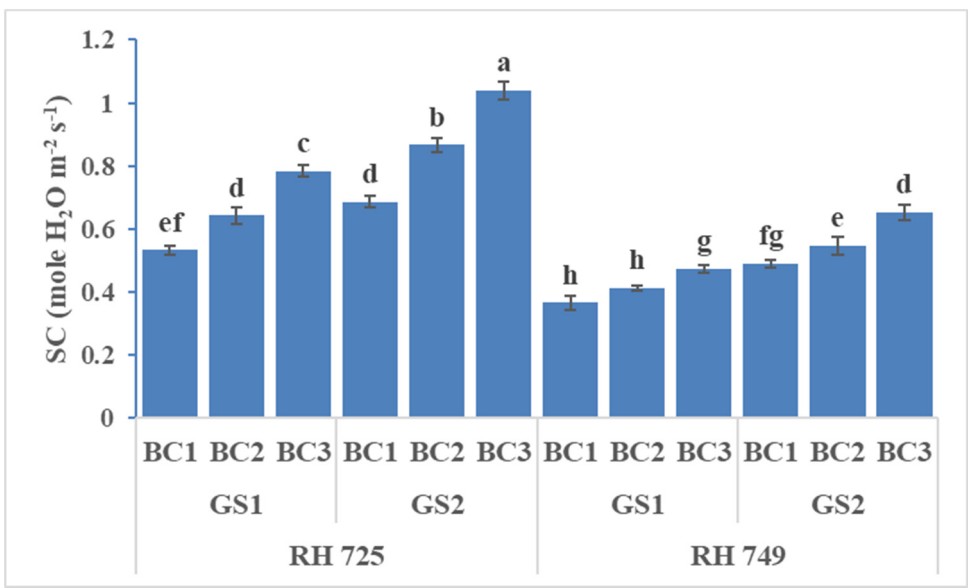

**Figure 1.** Comparisons of the effects of different concentrations of brassinolide sprays (BC1: control (water spray); BC2: 10 mg BRs/L water; BC3: 20 mg BRs/L water) on the stomatal conductance (SC) of drought-tolerant (RH 725) and drought-sensitive (RH 749) Indian mustard cultivars at the flower initiation (GS1) and 50% flowering (GS2) stages. Columns marked by different letters indicate significant differences ($p < 0.05$) between treatments based on Duncan's multiple range test. Error bars denote the standard errors of the mean.

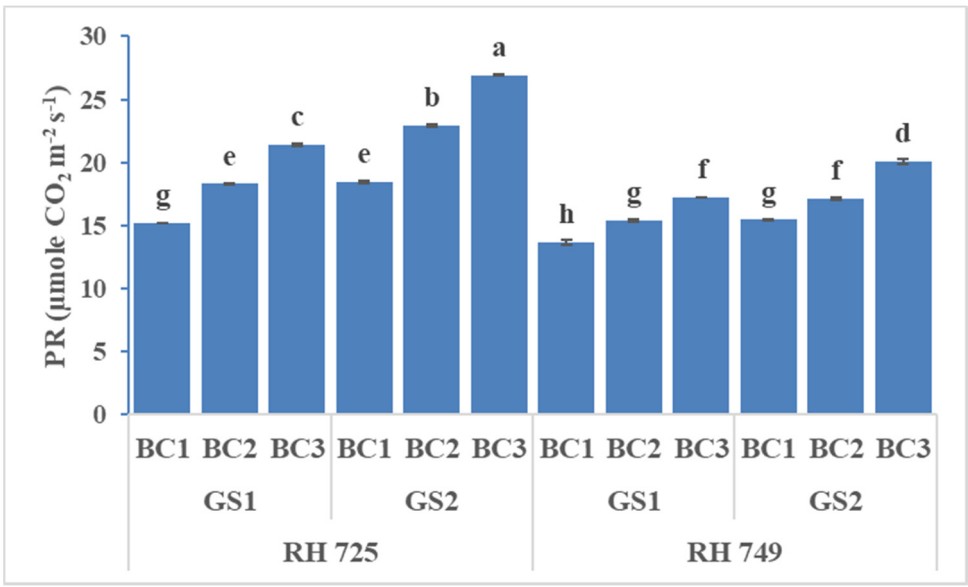

**Figure 2.** Comparisons of the effects of different concentrations of brassinolide sprays (BC1: control (water spray); BC2: 10 mg BRs/L water; BC3: 20 mg BRs/L water) on the photosynthetic rate (PR) of drought-tolerant (RH 725) and drought-sensitive (RH 749) Indian mustard cultivars at the flower initiation (GS1) and 50% flowering (GS2) stages. Columns marked by different letters indicate significant differences ($p < 0.05$) between treatments based on Duncan's multiple range test. Error bars denote the standard errors of the mean.

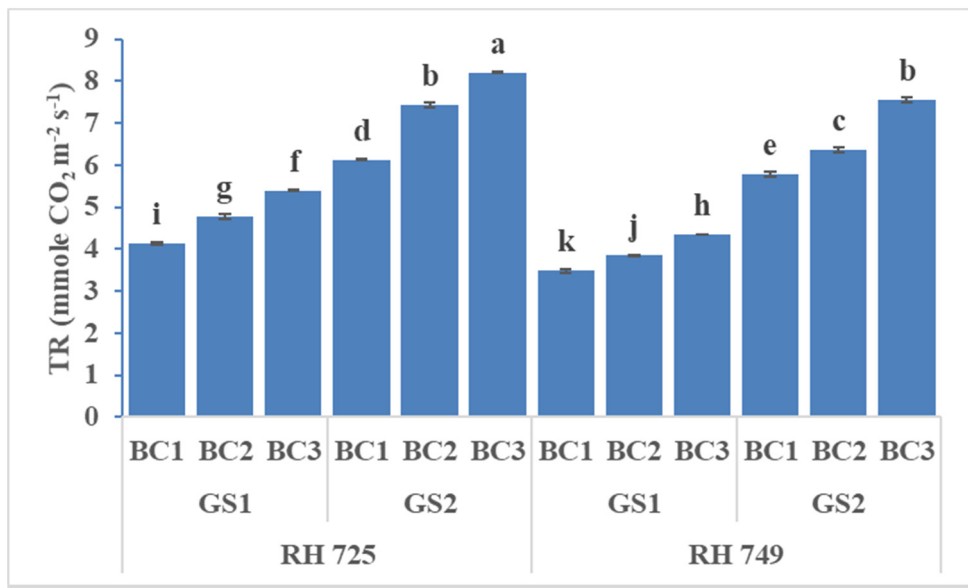

**Figure 3.** Comparisons of the effects of different concentrations of brassinolide sprays (BC1: control (water spray); BC2: 10 mg BRs/L water; BC3: 20 mg BRs/L water) on the transpiration rate (TR) of drought-tolerant (RH 725) and drought-sensitive (RH 749) Indian mustard cultivars at the flower initiation (GS1) and 50% flowering (GS2) stages. Columns marked by different letters indicate significant differences ($p < 0.05$) between treatments based on Duncan's multiple range test. Error bars denote the standard errors of the mean.

### 3.2. Oxidative Stress Indicators

Three-way ANOVA (Table 1) showed highly significant effects of all the three individual factors and their interactions on oxidative stress indicators, except that C × BC and C × GS × BC were insignificant for malondialdehyde, while GS × BC and C × GS × BC were insignificant for electrolyte leakage. Figures 4–6 show that brassinolide treatments significantly decreased the oxidative stress indicators—i.e., hydrogen peroxide ($H_2O_2$), malondialdehyde (MDA), and electrolyte leakage (EL)—in Indian mustard. Brassinolide foliar application at 20 mg/L showed a maximum decrease in $H_2O_2$, MDA, and EL over their respective controls at the 50% flowering stage of growth in RH 725. The percentage decrease in $H_2O_2$, MDA, and EL caused by brassinolide (20 mg/L) was 32.14, 60.37, and 37.94%, respectively, in RH 725 at the 50% flowering stage. On the other hand, at the same plant growth stage, this decrease was 18.56%, 54.98%, and 20.68% in $H_2O_2$, MDA, and EL, respectively, at 20 mg/L in RH 749, which was a sensitive cultivar. It is clear from these results that the tolerant cultivar RH 725 showed a significantly better response to brassinolide as compared to the sensitive cultivar RH 749.

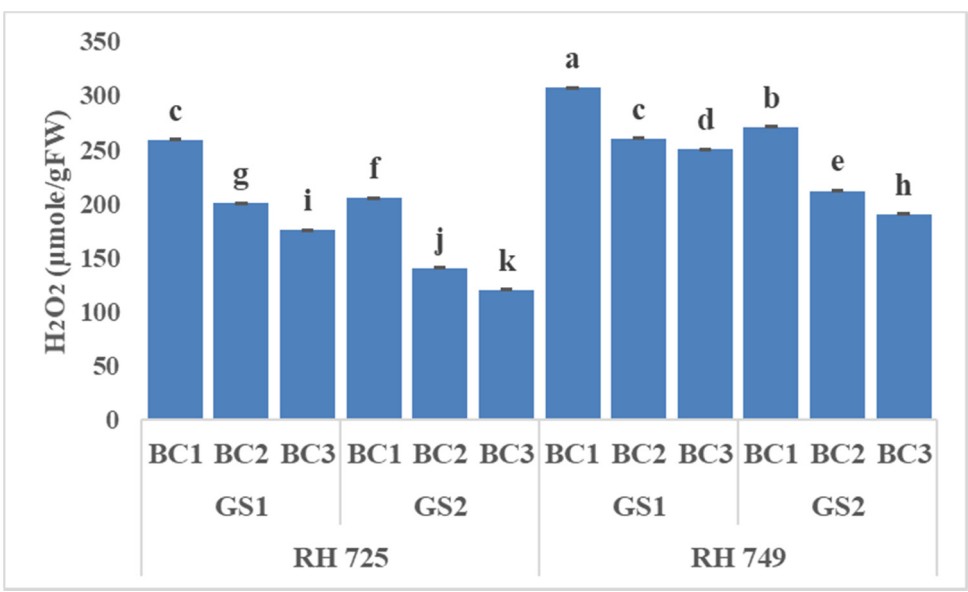

**Figure 4.** Comparisons of the effects of different concentrations of brassinolide sprays (BC1: control (water spray); BC2: 10 mg BRs/L water; BC3: 20 mg BRs/L water) on the hydrogen peroxidase ($H_2O_2$) concentration of drought-tolerant (RH 725) and drought-sensitive (RH 749) Indian mustard cultivars at the flower initiation (GS1) and 50% flowering (GS2) stages. Columns marked by different letters indicate significant differences ($p < 0.05$) between treatments based on Duncan's multiple range test. Error bars denote the standard errors of the mean.

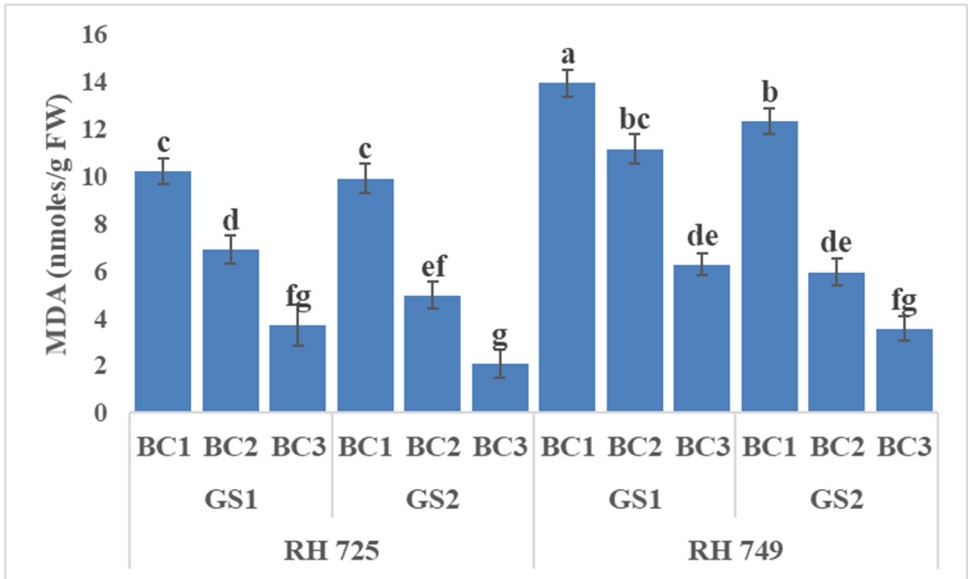

**Figure 5.** Comparisons of the effects of different concentrations of brassinolide sprays (BC1: control (water spray); BC2: 10 mg BRs/L water; BC3: 20 mg BRs/L water) on the malondialdehyde (MDA) concentration of drought-tolerant (RH 725) and drought-sensitive (RH 749) Indian mustard cultivars at the flower initiation (GS1) and 50% flowering (GS2) stages. Columns marked by different letters indicate significant differences ($p < 0.05$) between treatments based on Duncan's multiple range test. Error bars denote the standard errors of the mean.

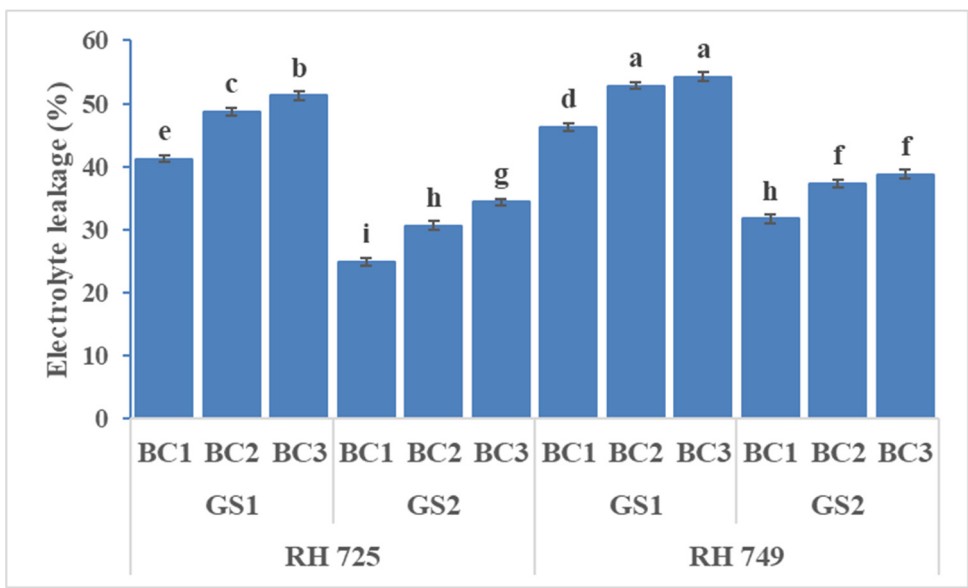

**Figure 6.** Comparisons of the effects of different concentrations of brassinolide sprays (BC1: control (water spray); BC2: 10 mg BRs/L water; BC3: 20 mg BRs/L water) on the electrolyte leakage (EL) concentration of drought-tolerant (RH 725) and drought-sensitive (RH 749) Indian mustard cultivars at the flower initiation (GS1) and 50% flowering (GS2) stages. Columns marked by different letters indicate significant differences ($p < 0.05$) between treatments based on Duncan's multiple range test. Error bars denote the standard errors of the mean.

### 3.3. Enzymatic Antioxidants

Three-way ANOVA revealed statistical significant effects of cultivar (C), growth stage (GS), brassinolide concentration (BC), and their interaction on SOD and CAT, whereas only the individual factors had significant effects on POX (Table 1). Brassinolide at two concentrations (10 and 20 mg/L), when applied to the plant, enhanced the activity of antioxidative enzymes (SOD, CAT, and POX) to a significant extent, but the tolerant cultivar (RH 725) showed significantly higher ($p < 0.05$) activities over the growth stages. However, all of the enzyme activities were increased significantly—particularly at the 50% flowering stage. The tolerant cultivar RH 725 exhibited the highest enzymatic activities (SOD, POX, and CAT) as compared to the sensitive cultivar RH 749 at the 50% flowering stage. Brassinolide (20 mg/L) significantly enhanced the SOD, CAT, and POX activity by 31.73, 22.80, and 45.07% at the flower initiation stage and 36.92, 27.86, and 48.12% at the 50% flowering stage, respectively, in comparison to their controls in drought-tolerant RH 725, while this increase was less in the drought-sensitive cultivar RH 749 (Figures 7–9).

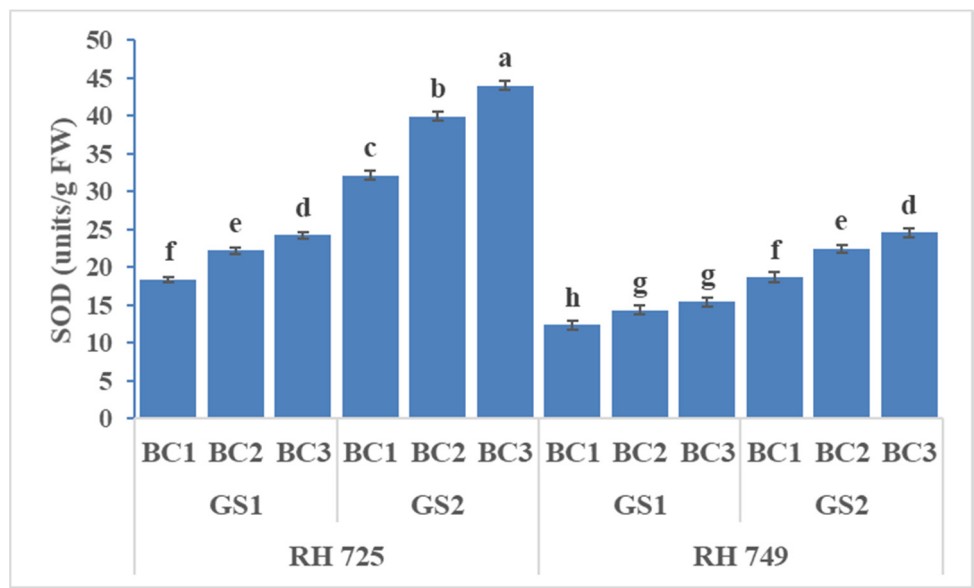

**Figure 7.** Comparisons of the effects of different concentrations of brassinolide sprays (BC1: control (water spray); BC2: 10 mg BRs/L water; BC3: 20 mg BRs/L water) on the superoxide dismutase (SOD) activity of drought-tolerant (RH 725) and drought-sensitive (RH 749) Indian mustard cultivars at the flower initiation (GS1) and 50% flowering (GS2) stages. Columns marked by different letters indicate significant differences ($p < 0.05$) between treatments based on Duncan's multiple range test. Error bars denote the standard errors of the mean.

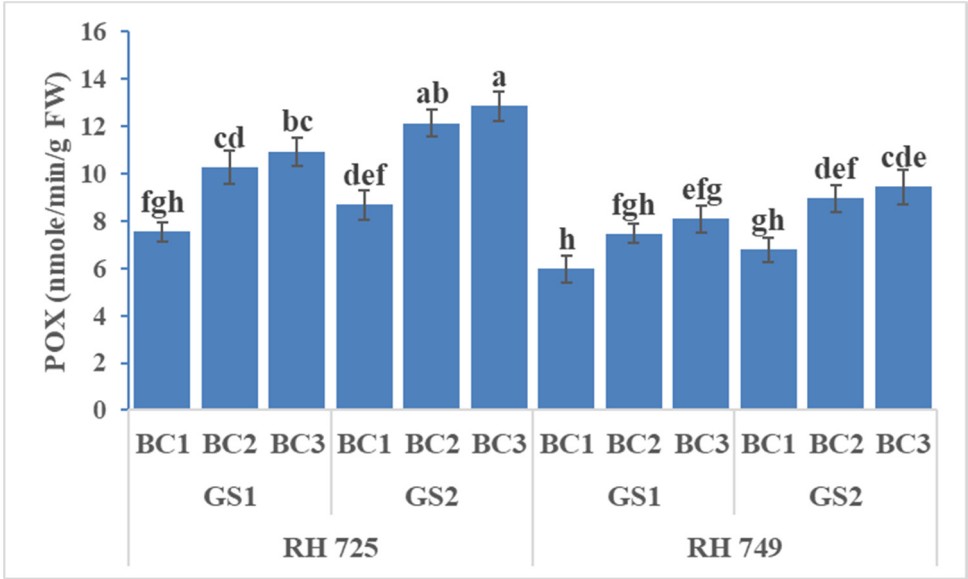

**Figure 8.** Comparisons of the effects of different concentrations of brassinolide sprays (BC1: control (water spray); BC2: 10 mg BRs/L water; BC3: 20 mg BRs/L water) on the peroxidase (POX) activity of drought-tolerant (RH 725) and drought-sensitive (RH 749) Indian mustard cultivars at the flower initiation (GS1) and 50% flowering (GS2) stages. Columns marked by different letters indicate significant differences ($p < 0.05$) between treatments based on Duncan's multiple range test. Error bars denote the standard errors of the mean.

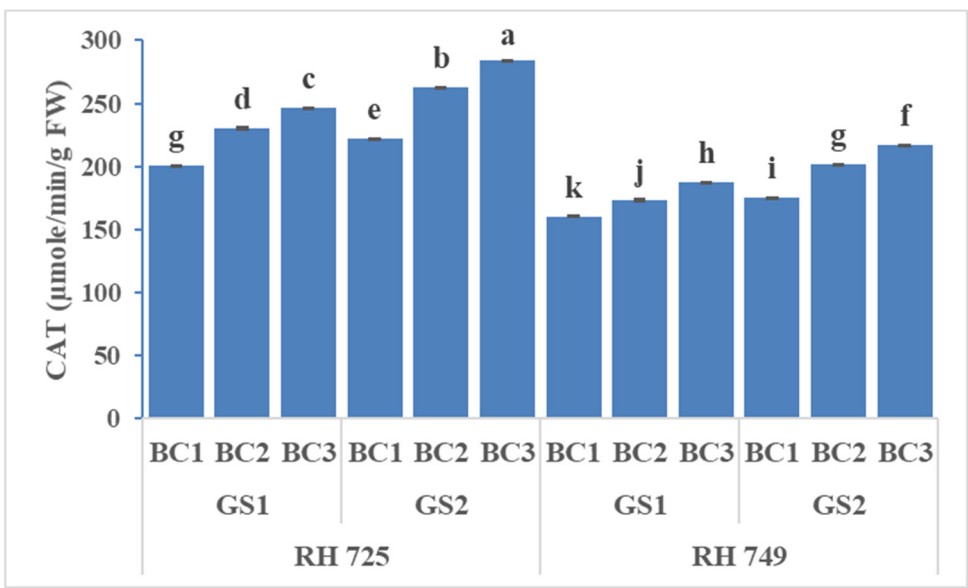

**Figure 9.** Comparisons of the effects of different concentrations of brassinolide sprays (BC1: control (water spray); BC2: 10 mg BRs/L water; BC3: 20 mg BRs/L water) on the catalase (CAT) activity of drought-tolerant (RH 725) and drought-sensitive (RH 749) Indian mustard cultivars at the flower initiation (GS1) and 50% flowering (GS2) stages. Columns marked by different letters indicate significant differences ($p < 0.05$) between treatments based on Duncan's multiple range test. Error bars denote the standard errors of the mean.

### 3.4. Non-Enzymatic Antioxidants

ANOVA showed significant effects of cultivar, growth stage, and brassinolide concentration on all three non-enzymatic antioxidants, while their interactions—viz.,C × GS, C × BC, and GS × BC—were significant for ascorbic acid only (Table 1). It is evident from Figures 10–12 that foliar application of brassinolide (10 and20 mg/L) significantly increased the non-enzymatic attributes in both cultivars, but this increase was more pronounced in the tolerant cultivar RH 725 as compared to RH 749,with increases in the levels of carotenoids, ascorbic acid, and proline (at 20 mg/L)of 27.04, 49.63, and 40.91% at the flower initiation stage and 41.55, 64.11, and 45.22% at the 50% flowering stage, respectively, over their respective controls in RH 725. Of the two concentrations of brassinolide and two stages of plant growth studied, 20 mg/L of brassinolide and the 50% flowering stage showed the greatest response in RH 725; this increase was less significant in the sensitive cultivar RH 749.

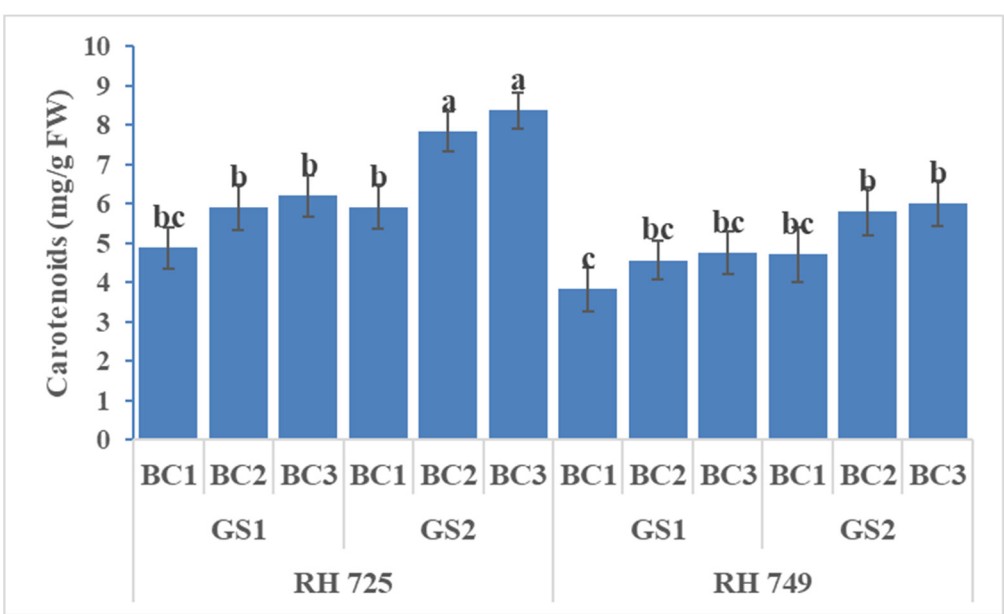

**Figure 10.** Comparisons of the effects of different concentrations of brassinolide sprays (BC1: control (water spray); BC2: 10 mg BRs/L water; BC3: 20 mg BRs/L water) on the carotenoids content (CC) of drought-tolerant (RH 725) and drought-sensitive (RH 749) Indian mustard cultivars at the flower initiation (GS1) and 50% flowering (GS2) stages. Columns marked by different letters indicate significant differences ($p < 0.05$) between treatments based on Duncan's multiple range test. Error bars denote the standard errors of the mean.

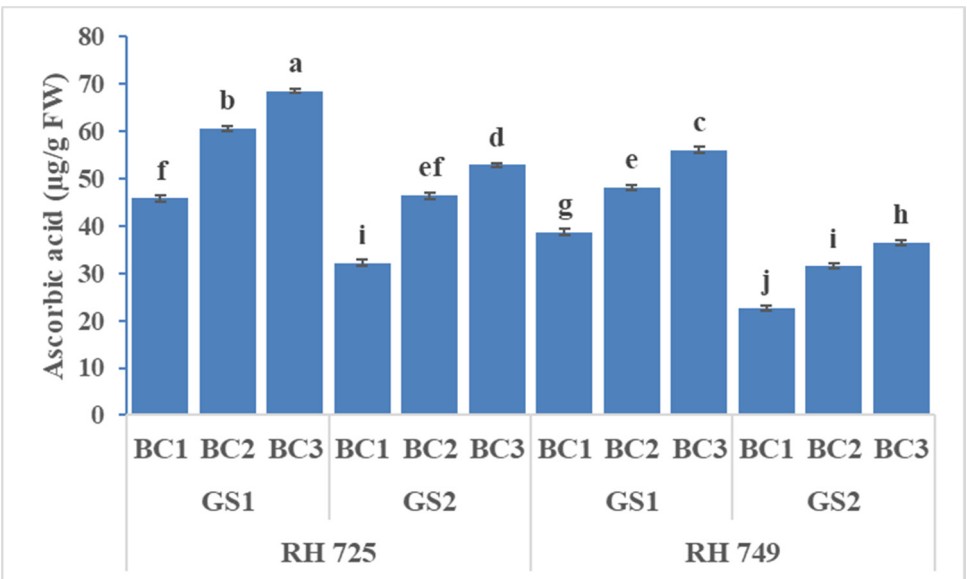

**Figure 11.** Comparisons of the effects of different concentrations of brassinolide sprays (BC1: control (water spray); BC2: 10 mg BRs/L water; BC3: 20 mg BRs/L water) on the ascorbic acid content (ASA) of drought-tolerant (RH 725) and drought-sensitive (RH 749) Indian mustard cultivars at the flower initiation (GS1) and 50% flowering (GS2) stages. Columns marked by different letters indicate significant differences ($p < 0.05$) between treatments based on Duncan's multiple range test. Error bars denote the standard errors of the mean.

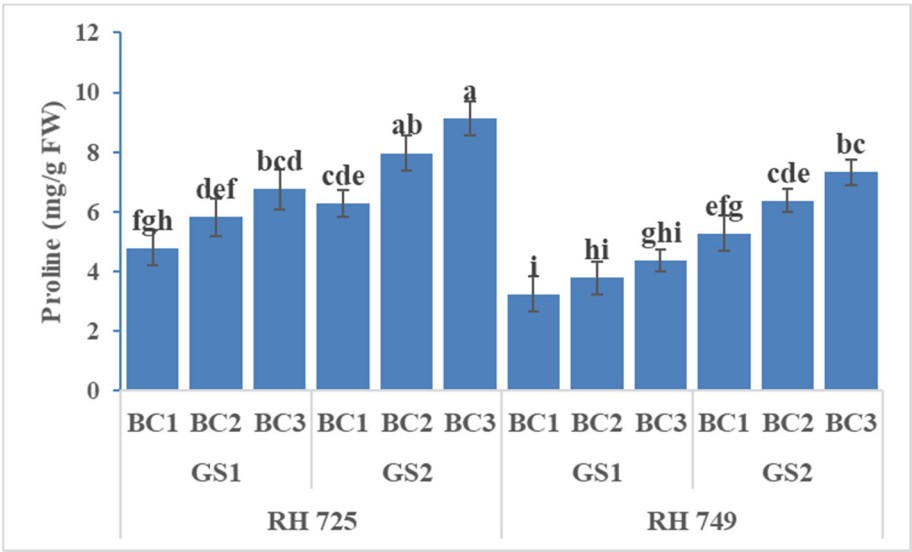

**Figure 12.** Comparisons of the effects of different concentrations of brassinolide sprays (BC1: control (water spray); BC2: 10 mg BRs/L water; BC3: 20 mg BRs/L water) on the proline content (PRO) of drought-tolerant (RH 725) and drought-sensitive (RH 749) Indian mustard cultivars at the flower initiation (GS1) and 50% flowering (GS2) stages. Columns marked by different letters indicate significant differences ($p < 0.05$) between treatments based on Duncan's multiple range test. Error bars denote the standard errors of the mean.

### 3.5. Correlation Analysis among Different Parameters

The Pearson's correlation coefficient matrix presented in Table 2 reveals the significant negative associations between both enzymatic and non-enzymatic antioxidants and oxidative stress indicators—*viz.*, $H_2O_2$ and MDA—except for ascorbic acid, which showed insignificant association between $H_2O_2$ and MDA, while it was significant and positively correlated with EL. Moreover, both of the oxidative stress indicators—$H_2O_2$ and MDA—showed a significant positive relationship with one another. The EL, which is the most important oxidative stress indicator, showed a significant negative association with SOD only. Proline content and all physiological parameters—i.e., stomatal conductance, photosynthetic rate, and transpiration rate—were positively correlated with all of the enzymatic antioxidants, while they were negatively associated with all oxidative stress indicators except for electrolyte leakage, which showed a significant negative association with transpiration rate only.

**Table 2.** Pearson's product–moment correlation matrix between different physio-biochemical parameters evaluated during the present study.

| Variables | SOD | POX | CAT | CC | ASA | PRO | $H_2O_2$ | MDA | EL | SC | PR | TR |
|---|---|---|---|---|---|---|---|---|---|---|---|---|
| SOD | 1.000 | 0.864 ** | 0.911 ** | 0.960 ** | 0.091 | 0.920 ** | −0.909 ** | −0.644* | −0.640 * | 0.948 ** | 0.903 ** | 0.857 ** |
| POX | | 1.000 | 0.974 ** | 0.962 ** | 0.494 | 0.901 ** | −0.981 ** | −0.873 ** | −0.220 | 0.945 ** | 0.955 ** | 0.739 ** |
| CAT | | | 1.000 | 0.963 ** | 0.442 | 0.916 ** | −0.969 ** | −0.811 ** | −0.338 | 0.983 ** | 0.950 ** | 0.743 ** |
| CC | | | | 1.000 | 0.263 | 0.957 ** | −0.978 ** | −0.801 ** | −0.455 | 0.970 ** | 0.959 ** | 0.856 ** |
| ASA | | | | | 1.000 | 0.118 | −0.355 | −0.481 | 0.657 * | 0.332 | 0.388 ** | −0.172 |
| PRO | | | | | | 1.000 | −0.953 ** | −0.824 ** | −0.537 | 0.941 ** | 0.927 ** | 0.943 ** |
| $H_2O_2$ | | | | | | | 1.000 | 0.874 ** | 0.353 | −0.957 ** | −0.960 ** | −0.832 ** |
| MDA | | | | | | | | 1.000 | 0.025 | −0.774 ** | −0.854 ** | −0.711 ** |
| EL | | | | | | | | | 1.000 | −0.434 | −0.304 | −0.666 ** |
| SC | | | | | | | | | | 1.000 | 0.970 ** | 0.806 ** |
| PR | | | | | | | | | | | 1.000 | 0.817 ** |
| TR | | | | | | | | | | | | 1.000 |

** Significant at $p \leq 0.01$; * significant at $p \leq 0.05$; SOD: superoxide dismutase; PX: peroxidase; CAT: catalase; CC: carotenoids content; ASA: ascorbic acid content; PRO: proline content; $H_2O_2$: hydrogen peroxide content; MDA: malondialdehyde content; EL: electrolyte leakage; SC: stomatal conductance; PR: photosynthetic rate; TR: transpiration rate.

## 4. Discussion

Analysis of variance showed significant effects of cultivar, growth stage, brassinolide, and their interactions on most of the studied traits, indicating that drought tolerance in Indian mustard is a cultivar- and growth-stage-specific but brassinolide-responsive trait. Similar patterns of results were obtained in many previous studies [31–35]. This indicates that the drought-tolerant cultivar RH 725 is more responsive to brassinolide, and has the capacity to cope with drought-induced oxidative stress and detoxify the oxidative stress indicators by significantly elevating both the non-enzymatic and enzymatic antioxidants. Nevertheless, drought-tolerant RH 725 also has the capacity to improve the physiological processes—*viz.*, stomatal conductance, photosynthetic rate, and transpiration rate—in order to maintain better physiology of the plant as compared to sensitive RH 749. The foliar application of brassinolide enhanced the levels of both enzymatic and non-enzymatic antioxidants, and caused decreased production of oxidative stress indicators (MDA, $H_2O_2$, and EL). However, we found that the physio-biochemical contents/activities in the leaves depend on the genotype of the cultivar. The drought-tolerant cultivar RH 725 had significantly higher levels of both enzymatic and non-enzymatic antioxidants, and higher physiological parameters, along with lower levels of oxidative stress indicators as compared to the drought-sensitive cultivar RH 749. This indicates that the drought-tolerant cultivar—particularly at the 50% flowering stage—is more responsive to exogenous application of BRs in terms of mitigating drought stress, compared to the sensitive cultivar. Similar results were also reported previously in many crops, including maize [36], sunflowers, [37]; tomatoes, [38], and chickpeas [39].

Brassinosteroids are attractive as original regulators in plants because of their ability to enhance cells in two ways: to provide defense, and to promote growth [40]. Tolerance provided by BR treatment is mediated via the provoked expression of genes involved in defense, regulation, antioxidant responses, and the production of high levels of $H_2O_2$,which results from enhanced activity of NADPH oxidase [41]. Brassinosteroids regulate the activity of antioxidative enzymes in the cells where ROS production is very high [42]. These results are consistent with the findings of Behnamnia et al. [18], who reported significant augmentation in SOD, CAT, and POX activity in *Lycopersicon esculentum* with the application of brassinolide under drought stress. Similar effects of brasinosteroids were also observed in maize [43], soybeans [44], wheat [45], and Indian mustard [16]. These findings consistent with those of Kumari and Thakur [46], who reported that BRs could regulate antioxidant enzymes such as superoxide dismutase, catalase, peroxidase, etc., in plants under different stress conditions.

Non-enzymatic antioxidants such as carotenoids, ascorbic acid, and proline play a vital rolein the metabolism of plants, by shielding them from stress conditions [47]. Plants produce the carotenoids, which are natural pigments and are involved in photoprotection and photosynthesis. Under drought conditions, carotenoids increase significantly. Ascorbic acid is one of the most powerful antioxidants, which scavenge harmful free radicals and other ROS [48]. Brassinolide was reported to increase the contents of ascorbic acid and total carotenoids in seedlings of drought-resistant (PAN 6043) and drought-sensitive (SC 701) cultivars of *Zea mays* under water stress [43]. Plants accrue low-molecular-mass compounds, such as proline [49], which acts as a non-enzymatic antioxidant that is well known to stabilize the sub-cellular structures of proteins and cell membranes, scavenging free radicals and buffering redox potential under various stress conditions. Proline also acts as a molecular chaperone that preserves the integrity of proteins and boosts the activity of various enzymes during stressful conditions [50]. Among the different compatible solutes, proline is the only molecule that protects the plants against singlet oxygen and damage induced by free radicals resulting from various stresses [51]. It has also been reported previously that BRs propel the expression of proline biosynthetic genes [52]. High proline content in plants under water stress is frequently observed in several plant species [53,54].

The product of membrane peroxidation is the thiobarbituric-acid-reactive substance malondialdehyde (MDA),which is used as a direct marker of membrane damage and lipid

peroxidation. Reactive oxygen species attack the majority of the sensitive macromolecules in cells under various environmental stresses, interfering with their function. Drought stress resulted in an increase in MDA accumulation in the leaves of Indian mustard [53]. It was reported that level of lipid peroxidation induced by biotic stresses—such as oxygen deficiency [55], drought stress [56], and heat [57]—could be decreased by treatment with BRs. The results of the present study are consistent with earlier findings that the level of lipid peroxidation in *Brassica juncea* leaves was augmented during drought stress, and it was significantly minimized by BR application. Hydrogen peroxide is produced in the cells under normal as well as a wide range of stressful conditions, such as drought, chilling, UV irradiation, exposure to intense light, wounding, and intrusion by pathogens; it can generate singlet oxygen upon reaction with superoxide anions/HOCl, and it can degrade certain heme proteins to release iron ions, so it is considered to increase membrane permeability by degrading membrane lipids [58]. Therefore, it is important that $H_2O_2$ be scavenged rapidly by the antioxidative defense system.

Leaf membrane damage was determined by measurement of electrolyte leakage (EL), as described by Valentovic et al. [59]. Electrolyte leakage decreased considerably in *Curcuma alismatifolia* when subjected to water-deficit stress [60]. This is an indicator of a drought-tolerance mechanism in the species via the maintenance of membrane integrity and reduction in electrolyte leakage. The exposure of the plants to drought stress resulted in an increase in electrolyte leakage, which was mitigated by spraying with brassinolide. Houimli et al. [61] observed that exogenous application of brassinolide resulted in a significant reduction in electrolyte leakage under salt stress. Similarly, Coban and Baydar [62] reported a significant reduction in electrolyte leakage in maize, along with improved morphometric parameters, when brassinolide was applied.

Stomatal conductance, photosynthetic rate, and transpiration rate are important characteristics describing plants' water relations [63]. These are inter-related traits of plants that play major roles under stress conditions. In the present study, there were significant increases in stomatal conductance, photosynthetic rate, and transpiration rate with brassinolide spraying under drought stress. Similar increases in these parameters due to BR application have previously been observed in tomatoes, wheat, and cucumbers under both normal conditions and environmental stresses [41,45,57]. Brassinosteroids are also known to activate the key enzymes of photosynthesis, i.e., rubisco [64] and carbonic anhydrase [65]. The assimilation of $CO_2$ in the Calvin cycle is increased by high carbonic anhydrase activity, which is primarily ascribed to efficient functioning of rubisco [66], consequently improving the net photosynthetic rate and related attributes.

Furthermore, the significant negative association of both enzymatic and non-enzymatic antioxidants with oxidative stress indicators, along with their positive correlation with physiological parameters, again confirmed the role of the antioxidant defense system in mitigating the negative effects of drought stress. However, a further analysis of this phenomenon is certainly needed.

## 5. Conclusions

The present investigation found that both concentrations of brassinolide (10 and 20 mg/L) improved the plants' efficiency via different physio-biochemical amendments. However, with the 20 mg/L brassinolide spray at the 50% flowering stage, various physio-biochemical attributes showed a more emphatic response in RH 725 than in RH 749. Enhancement of the antioxidative system with improved antioxidative enzyme activity and accumulation of proline may strengthen the plants' ability to combat different stress conditions. Moreover, the drought-tolerant cultivar (RH 725) was superior in term of antioxidant defense system, as compared with the sensitive cultivar RH 749. Indeed, this could be one of the reasons for the former's higher drought tolerance. Understanding the mechanisms of drought tolerance in Indian mustard will make it possible for plant breeders and plant physiologists to develop specific techniques to mitigate the adverse effects of drought, and to maximize Indian mustard crop production.

**Supplementary Materials:** The following are available online at https://www.mdpi.com/article/10.3390/horticulturae7110514/s1, Supplementary Table S1: Weekly averaged rainfall data during the crop season of 2018-19 at CCS Haryana Agricultural University, Hisar.

**Author Contributions:** Conceptualization, N.K. and R.A.; methodology, N.K., R.A. and B.R.; validation, M.J., S.A. and N.K.; software, M.S., resources, R.A., investigation, N.N. and N.K.; data curation, M.S., N.N. and K.M.; writing—original draft, N.N.; writing—review and editing, M.S., N.K. and A.S. All authors have read and agreed to the published version of the manuscript.

**Funding:** This research was supported by Oilseeds Section, Department of Genetics and Plant Breeding, CCS Haryana Agricultural University, Hisar, India. [C (b) PB-3-ICAR].

**Institutional Review Board Statement:** Not applicable.

**Informed Consent Statement:** Not applicable.

**Data Availability Statement:** The data used for the analysis in this study are available within the article, while the datasets used or analyzed during the current study are available from the corresponding author upon reasonable request.

**Acknowledgments:** The first author is highly grateful to HOS, Oilseeds Section, Department of Genetics and Plant Breeding, CCS Haryana Agricultural University, Hisar, for providing the necessary facilities throughout her M.Sc. research work.

**Conflicts of Interest:** The authors declare no conflict of interest.

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
