# Peer review of "Evaluation of Effect of Brassinolide in Brassica juncea Leaves under Drought Stress in Field Conditions"

_horticulturae, doi:10.3390/horticulturae7110514_

Round 1

Reviewer 1 Report

Very well written, the results are clearly presented, enjoyed reading the manuscript.

 I have a few questions:

  1. . Does this research fit the topic" Brassica Crops Genomics and Breeding"?
  2. What is the purpose of selecting the two varieties RH-725 and RH-749?
  3. Drought treatment were achieved by withholding irrigation, how to control the soil moisture affected by rainfall in field condition?
  4. Brassinolid treatment enhanced the photosynthetic rate, does the enhancement affect seeds yield?
  5. All figures are needed to show the standard deviation error bar, and the statistical significance test.
  6. Need more explaining of the result of Table 5. How were the results obtained?

Author Response

Response to Reviewer reports

Manuscript ID: horticulturae-1392032: “Evaluating the Effect of Brassinolide in Brassica Juncea Leaves Under Drought Stress at Field Conditions.”

We are thankful to the Editor and the reviewers for their keen observations and valuable suggestions regarding the manuscript. We have now revised the manuscript as per the suggestions and point-by-point response to the queries is appended below.

Reviewer #1

Comment 1: Does this research fit the topic" Brassica Crops Genomics and Breeding"?

Response:

I thank the learned reviewer for critically evaluating our work. This manuscript was submitted as per editor’s call in this special issue. Besides, this study provides initial information about the further investigation of Brassica crop genomics and breeding for drought tolerance.

Comment 2: What is the purpose of selecting the two varieties RH-725 and RH-749?

Response:

These varieties exhibit contrasting response to drought stress. RH 725 is drought tolerant cultivar and recommended for cultivation under rainfed condition while RH 749 is a drought sensitive cultivar and recommended for cultivation under irrigated conditions. Therefore, these varieties were selected on the basis of their differential response to drought.

Comment 3: Drought treatment was achieved by withholding irrigation, how to control the soil moisture affected by rainfall in field condition?

Response:

The amount of rainfall was negligible during the crop period and the effect of drought was quite clear in the present investigation (Please refer supplementary Table 1 about the rainfall data during the crop grown period).

Comment 4: Brassinolide treatment enhanced the photosynthetic rate, does the enhancement affect seeds yield?

Response:

Yes, besides enhancing seed yield, brassinolide treatment also significantly elevates the oil content in both varieties but the seed and oil yield enhancement were pronounced in tolerant cultivar RH 725 as compared to sensitive cultivar RH 749. The effects of brassinolide on seed yield/plant, 1000-seed weight (g) and oil content (%) of drought tolerant RH 725 and drought sensitive RH 749 is presented below:

Seed yield/plant

1000 seed weight (g)

Oil-content (%)

Treatments

RH-725

RH-749

RH-725

RH-749

RH-725

RH-749

Control

22.8

22.5

6.6

6.3

39.5

39.7

10ppm

25.5

(11.84)

23.8

(5.77)

7.4

(12.12)

6.6

(4.76)

40.0

(1.26)

40.1

(1.00)

20ppm

26.2

(14.91)

24.2

(4.55)

7.6

(15.15)

6.9

(9.52)

41.4

(4.810)

40.3

(1.51)

CD  (5%)

Variety =0.296

Treatment=0.362

Variety × treatment=0.384

Variety =0.224    

Treatment=0.275

Variety × Treatment=N.S

Variety =N.S

Treatment=0.347

Variety × Treatment=0.491

*Values in the parenthesis is +/- in percentage over the control

Comment 5: All figures are needed to show the standard deviation error bar, and the statistical significance test.

Response:

As per reviewer kind suggestions, changes have been incorporated in the manuscript (please see the Results section).

Comment 5: Need more explaining of the result of Table 5. How were the results obtained?

Response:

As per reviewer kind suggestions, changes have been incorporated in the manuscript (please see results as well as discussion section). These results were obtained by Pearson’s product moment correlation analysis between different physio-biochemical parameters studied during the present investigation.

Reviewer 2 Report

Dear Authors,

I appreciate your research work submitted in this journal. However, I found critical problems in the manuscript as:

  1. The objective is not clear.
  2. Methods for measuring the physiological parameters (stomatal conductance, photosynthetic and transpiration rate) is missing.
  3. Tables and figure must be improve for better readability.
  4. Conclusion is not clear.

The other comments are:

  1. Line 91-94: Write clear objective.
  2. Line 98: provide the location of the experiment.
  3. The tables need to modify. You can see the recent publications in this journal.
  4. In figure 1, 2, and 3, control and 20 ppm have same color. It will be better to modify the color in legends. Furthermore, both the flower initiation and 50% flower stage graph have same legends. Separate these in x axis. In addition, all the sub sections (3 subsections) of each figure in a single row or column will be better.
  5. Please write supportive information in discussion. You have only discussed what the other researchers have found.
  6. Make a clear conclusion
  7. See in detail the author guidelines for reference section.
  8. There are several typo errors in the manuscript. Please check carefully.

With Regards

Reviewer

Author Response

Response to Reviewer reports

Manuscript ID: horticulturae-1392032: “Evaluating the Effect of Brassinolide in Brassica Juncea Leaves Under Drought Stress at Field Conditions.”

We are thankful to the Editor and the reviewers for their keen observations and valuable suggestions regarding the manuscript. We have now revised the manuscript as per the suggestions and point-by-point response to the queries is appended below.

Reviewer #2

Comment 1: The objective is not clear.

Response:

Changes has been incorporated in the manuscript file as per reviewer’ kind suggestion.

The objectives of the current study were: 1) whether exogenous application of brassinolide could alleviate drought stress in Indian mustard (2) whether the drought tolerant and drought sensitive varieties have similar response to these treatments under drought stress.

Comment 2: Methods for measuring the physiological parameters (stomatal conductance, photosynthetic and transpiration rate) are missing.

Response:

Changes has been incorporated in the manuscript file as per reviewer’ kind suggestion (please see Materials and Methods section PP. 3).

The parameters like photosynthetic rate, stomatal conductance and transpiration rate were measured by Infrared Gas Analyzer (IRGA) system (LI-COR USA Model LI6400) and their units are Photosynthetic rate (PR, μmole CO2 m-2 s-1), Stomatal Conductance (SC, mole H2O m-2 s-1) and Transpiration rate (TR, mmole H2O m-2 s-1).

Comment 3: Tables and figure must be improve for better readability.

Response:

Tables and figures have been improved as per reviewer’ kind suggestion and changes has been incorporated in the manuscript file.

Comment 4: Conclusion is not clear.

Response:

Proper conclusions have been added in the manuscript file as per reviewer’ kind suggestion (please see the conclusion part in the revised manuscript)

The present investigation concluded that both the concentrations of brassinolide (10 & 20 mg/L) improved the plant efficiency through different physio-biochemical amendment. However, the 20 mg/L of brassinolide spray at 50% flowering stage, various physio-biochemical attributes showed more emphatic response in RH 725 as compared to RH 749. Enhanced antioxidative system with improved antioxidative enzyme activity and accumulation of proline may strengthen the plants to combat different stress conditions. Besides, the drought-tolerant variety (RH 725) was superior in term of antioxidant defense system as compared with the sensitive variety RH 749. Indeed, this could be one of the reasons for its higher drought tolerance. Understanding the mechanisms of drought tolerance in Indian mustard will make it possible for plant breeder and plant physiologists to develop specific techniques to mitigate adverse effects of drought and to maximize Indian mustard crop production.

Comment 5: Line 91-94: Write clear objective.

Response:

As per reviewer kind suggestions, clear objectives of the present study have been incorporated in the revised manuscript.

The objectives of the current study were: 1) whether exogenous application of brassinolide could alleviate drought stress in Indian mustard (2) whether the drought tolerant and drought sensitive varieties have similar response to these treatments under drought stress.

Comment 6: Line 98: provide the location of the experiment.

Response:

Changes incorporated in the revised manuscript as per reviewer suggestions (please see the materials and methods section).

The present investigation was carried out at Research Farm and Oil Quality Laboratory of Oilseeds Section, Department of Genetics and Plant Breeding, CCS Haryana Agricultural University, Hisar which is situated at a latitude of 29° 10’ N and longitude of 75° 46’ E and altitude 215.2 m above main sea level and fall in semi-tropical region of western zone of India.

Comment 7: The tables need to modify. You can see the recent publications in this journal.

Response:

Tables were modified as per the reviewer suggestion and changes incorporated in the revised manuscript.

Comment 8: In figure 1, 2, and 3, control and 20 ppm have same color. It will be better to modify the color in legends. Furthermore, both the flower initiation and 50% flower stage graph have same legends. Separate these in x axis. In addition, all the sub sections (3 subsections) of each figure in a single row or column will be better.

Response:

Figures were appropriately modified as per reviewer suggestions and were added in the revised manuscript (please refer Results section in the revised manuscript).

Comment 9: Please write supportive information in discussion. You have only discussed what the other researchers have found.

Response:

Supportive information was discussed in the revised manuscript as per reviewer suggestions.

Comment 10: Make a clear conclusion.

Response:

A clear conclusion is written as per the reviewer suggestion and added to revised manuscript.

Comment 11: See in detail the author guidelines for reference section.

Response:

A change in the reference section has been made in the main file as per reviewer suggestions.

Comment 12: There are several typo errors in the manuscript. Please check carefully.

Response:

Changes incorporated in the main file as per reviewer suggestions.

Reviewer 3 Report

The manuscript entitled “Evaluating the effect of brassinolide in Brassica juncea leaves under drought stress at field conditions” described the physiological effects of external application of plant phytohormone brassinolide on leaves of two cultivars of Brassica juncea upon drought conditions. The sustainable agriculture in observed climatic changes are extremely topical, therefore every efforts aimed at improving the yield of agricultural plants is meaningful.

However, current manuscript is did not meet the publication requirements for showing new achievements, only confirmed the phenomena already described in a given species. The authors did not attempt to present the results in an interesting way, presenting only tables and partially citing the data from the table. Data tables are presented in jumbled order. Table 1 presenting analysis of variance adds nothing to the discussion or the results and had no impact on discussion. Tables 2, 3 and 4 can be replaced by boxplots for example. There is no element of the conclusion what is the added value of the manuscript or future perspective concerning B. junceae breeding and yield. In addition, the choice of both B. juncea cultivars is unclear, were both different in drought tolerance or other factors? The method of drought application in field conditions was also not described. The methods of measurement all physiological parameters should be abbreviated or at least the name of the method specified.

Author Response

Response to Reviewer reports

Manuscript ID: horticulturae-1392032: “Evaluating the Effect of Brassinolide in Brassica Juncea Leaves Under Drought Stress at Field Conditions.”

We are thankful to the Editor and the reviewers for their keen observations and valuable suggestions regarding the manuscript. We have now revised the manuscript as per the suggestions and point-by-point response to the queries is appended below.

Reviewer #3

Comment 1: Current manuscript did not meet the publication requirements for showing new achievements, only confirmed the phenomena already described in a given species.

Response:

I would like to thank the learned reviewer for critically evaluating our work. This study further strengthens the information regarding role of brassinolide in mitigating drought stress and provides important information regarding physio-biochemical basis of drought tolerance in Indian mustard. Moreover, this information is scanty in mustard crop.

Comment 2: The authors did not attempt to present the results in an interesting way, presenting only tables and partially citing the data from the table. Data tables are presented in jumbled order. 

Response:

Contents of the file have been changed as per reviewer suggestions.

Comment 3: Table 1 presenting analysis of variance adds nothing to the discussion or the results and had no impact on discussion. 

Response:

Analysis of variance is now discussed in the discussion section as per reviewer suggestions.

Analysis of variance showed significant effect of varieties, growth stages, brassinolide and their interactions for most of studied traits, indicates that drought tolerance in Indian mustard is a varietal and growth stages specific but brassinolide responsive trait. Nearly similar pattern of results were obtained in many previous studies [31-35].

Comment 4: Tables 2, 3 and 4 can be replaced by boxplots for example.

Response:

Table 2, 3, and 4 is replaced by appropriate graphs and added in the main file as per reviewer suggestions

Comment 5: There is no element of the conclusion what is the added value of the manuscript or future perspective concerning B. juncea breeding and yield. 

Response:

Future prospective of the present study has been added in the conclusion part of the revised manuscript as per reviewer suggestions.

Comment 6: In addition, the choice of both B. juncea cultivars is unclear, were both different in drought tolerance or other factors?

Response:

Cultivars were selected on the basis of their differential response to drought tolerance. RH 725 is a drought tolerant cultivar while RH 749 is drought sensitive cultivars. Appropriate changes have been made in the main file as per reviewer suggestions.

Comment 7: The method of drought application in field conditions was also not described. 

Response:

Drought application method is now fully described in the revised manuscript as per reviewer suggestions.

Comment 8: The methods of measurement all physiological parameters should be abbreviated or at least the name of the method specified.

Response:

Measurement of all the physiological parameters has been described in the revised manuscript (please refer Materials and methods section)

The parameters like photosynthetic rate, stomatal conductance and transpiration rate were measured by Infrared Gas Analyzer (IRGA) system (LI-COR USA Model LI6400) and their units are Photosynthetic rate (PR, μmole CO2 m-2 s-1), Stomatal Conductance (SC, mole H2O m-2 s-1) and Transpiration rate (TR, mmole H2O m-2 s-1).

Round 2

Reviewer 2 Report

Dear Authors,

You have revised the manuscript well.  Please provide the reference for the measurement of stomatal conductance and other physiological parameters if available or describe each in brief.

With Regards

Reviewer

Author Response

Response to Reviewer reports (Minor Revision)

Manuscript ID: horticulturae-1392032: “Evaluating the Effect of Brassinolide in Brassica Juncea Leaves Under Drought Stress at Field Conditions.”

We are thankful to the Editor and the reviewers for their keen observations and valuable suggestions regarding the manuscript. We have now revised the manuscript as per the suggestions and point-by-point response to the queries is appended below.

Reviewer #2

Comment 1: You have revised the manuscript well.  Please provide the reference for the measurement of stomatal conductance and other physiological parameters if available or describe each in brief.

Response:

We would like to thank reviewer #2 for their comments in the first round that helped to improve the manuscript as well as their further comments here. Changes has been incorporated in the manuscript file as per reviewer’ kind suggestion. The measurement of stomatal conductance and other physiological parameters was done as per method employed by Silva et al. [23].

Silva, F.V.D.F.; Mendes, B.D.S.; Rocha, M.D.S.; Brito, J.F.D.; Beltrão, N.E.D.M.; Sofiatti, V. Photosynthetic pigments and gas exchange in castor bean under conditions of above the optimal temperature and high CO2. Acta Sci. Agron. 2015, 37, 331-337.

Reviewer 3 Report

The corrections incorporated to revised manuscript are very impressive and make the manuscript more clear and informative. However, in my opinion there are still major points which should be corrected. The main discrepancies I found in tree-way ANOVA interpretation. First of all, in case of statistical test we can talk only about statistical significance not biological significance. Therefore, “ significance” as in line 170 or 181 should be  replaced by “statistical significance”.

The interactions showed by three-way ANOVA should be described more carefully as a lot of interesting information can be concluded from this analysis, for example interesting can be phenomena of low impact of SOD on variety effect in contrast to other enzymatic antioxidants and how this can be explain?

Low interaction as well as discrepancies in interactions can bring some information leading to important conclusions about priming effect of brassinolides on plant resistant. Especially interesting can be differences in variety effects. Some parameters as hydrogen peroxidase (H2O2) concentration drastically differs between both cultivars.

The both names cultivars and variety are used interchangeably, it should be unified. Cultivar is traditionally for breeding and variety is for natural variabilities.

There is still lack of information about the criteria of choice of both cultivars for experiment, do you have any evidences that RH 725 is drought-tolerant whereas RH 749 is drought-sensitive? Maybe some article in which both are described?

Author Response

Response to Reviewer reports (Minor revisions)

Manuscript ID: horticulturae-1392032: “Evaluating the Effect of Brassinolide in Brassica Juncea Leaves Under Drought Stress at Field Conditions.”

We are thankful to the Editor and the reviewers for their keen observations and valuable suggestions regarding the manuscript. We have now revised the manuscript as per the suggestions and point-by-point response to the queries is appended below.

Reviewer #3

Comment 1: The corrections incorporated to revised manuscript are very impressive and make the manuscript more clear and informative. However, in my opinion there are still major points which should be corrected. The main discrepancies I found in tree-way ANOVA interpretation. First of all, in case of statistical test we can talk only about statistical significance not biological significance. Therefore, “ significance” as in line 170 or 181 should be  replaced by “statistical significance”.

Response:

We would like to thank reviewer #3 for their comments in the first round that helped to improve the manuscript as well as their further comments here. As per your praiseworthy suggestion, the word “significance” has been replaced with “statistical significance” in the revised manuscript file.

Comment 2: The interactions showed by three-way ANOVA should be described more carefully as a lot of interesting information can be concluded from this analysis, for example interesting can be phenomena of low impact of SOD on variety effect in contrast to other enzymatic antioxidants and how this can be explain?

Response:

As per reviewer praiseworthy suggestions, the three-way ANOVA interactions have been described more carefully in the revised MS. However, it is clear from the Table 1 and Figure 7 that varieties (cultivars) had significant effect on SOD activity. SOD activity was significant highest in drought tolerant RH 725 as compared to drought sensitive RH 749 in all the treatments (of different growth stages and different brassinolide concentration).

As per reviewer praiseworthy suggestions, changes have been made in the revised MS.

This indicates that drought tolerant cultivar RH 725 has more responsive to brassinolide and have the capacity to cope up with drought induced oxidative stress to detoxifying the oxidative stress indicators by significantly elevating the both non-enzymatic and enzymatic antioxidants. Despite this, drought tolerant RH 725 also has the capacity to improve the physiological processes viz. stomatal conductance, photosynthetic and transpirational rate to maintain better physiology of the plant as compared to sensitive RH 749.

Comment 3: Low interaction as well as discrepancies in interactions can bring some information leading to important conclusions about priming effect of brassinolides on plant resistant. Especially interesting can be differences in variety effects. Some parameters as hydrogen peroxidase (H2O2) concentration drastically differ between both cultivars.

Response:

Three ways ANOVA revealed statistically significant effect of cultivars, growth stages and brassinolide concentrations on H2O2 concentration (Table 1). It is cleared from Figure 4 that H2O2 concentration was significantly highest in drought sensitive RH 749 as compared to drought tolerant RH 725 in all the treatment which reveals the considerable effect of cultivars on H2O2 (the important oxidative indicator). However, increase in the brassinolide concentration helps in decreasing the H2O2 concentration but this decrease was significantly more in drought tolerant RH 725 as compared to sensitive RH 749. This indicates that drought tolerant cultivar is more responsive to brassinolide and has the capacity to cope up with the drought induced oxidative stress by significantly lower accumulation of H2O2.

As per reviewer praiseworthy suggestions, changes have been made in the revised MS.

This indicates that drought tolerant cultivar RH 725 has more responsive to brassinolide and have the capacity to cope up with drought induced oxidative stress to detoxifying the oxidative stress indicators by significantly elevating the both non-enzymatic and enzymatic antioxidants. Despite this, drought tolerant RH 725 also has the capacity to improve the physiological processes viz. stomatal conductance, photosynthetic and transpirational rate to maintain better physiology of the plant as compared to sensitive RH 749.

Comment 4: The both names cultivars and variety are used interchangeably, it should be unified. Cultivar is traditionally for breeding and variety is for natural variabilities.

Response:

The name “variety” has been replaced with “cultivar” in the revised MS.

Comment 5: There is still lack of information about the criteria of choice of both cultivars for experiment, do you have any evidences that RH 725 is drought-tolerant whereas RH 749 is drought-sensitive? Maybe some article in which both are described?

Response:

Both the cultivars were developed by our centre (Oilseeds Section, Department of Genetics and Plant Breeding, CCS Haryana Agricultural University, Hisar (India). The cultivar RH 725 is recommended for cultivation under rainfed conditions and hence possess tolerance to drought while the cultivar RH 749 is recommended is cultivation under irrigated conditions but not for rainfed condition and hence comparatively sensitive to drought. The evidence about both the cultivars is given in the separately attached PDF as supplementary file (Varietal description published by Directorate of Research, CCS HAU Hisar). Besides this, the package and practices for Rabi crop published by CCS HAU, Hisar and Annual Progress Report (2021) All India Coordinated Research Project on Rapeseed-Mustard, ICAR-Directorate of Rapeseed-Mustard Research, Bharatpur also provides evidence about these varieties and their water response (RH 725 as a rainfed cultivar and drought tolerant while RH 749 as a irrigated condition cultivar and drought sensitive).

References Link:

https://www.hau.ac.in/storage/app/uploads/NiJmbB1ETFnjIlnMxFONcSd41erdMftx8Sq8k2ZJ.pdf
